# Low prevalence of relative age effects in Luxembourg's male and female youth football

**Claude Simon[1], Fraser Carson[1,2], Irene Renate Faber[3,4], Thorben Hülsdünker[1,2]\***

**1** Department of Exercise and Sport Science, LUNEX International University of Health, Exercise and Sports, Differdange, Luxembourg, **2** Luxembourg Health & Sport Sciences Research Institute A.s.b.l., Differdange, Luxembourg, **3** Institute of Sport Science, University of Oldenburg, Oldenburg, Germany, **4** Research Centre Human Movement and Education, Windesheim University of Applied Sciences, Zwolle, The Netherlands

\* Thorben.huelsduenker@lunex-university.net

**Data Availability Statement:** This study used a third-party dataset of male and female junior soccer players owned by the Luxembourgish Football Federation (FLF). The FLF needs to be

## Abstract

The relative age effect (RAE) is a well-established phenomenon in football. However, while the majority of previous studies focussed on established football nations, it remains unclear if the constraint of a limited population of soccer players in smaller countries associated with less strict selection procedures may reduce the risk of RAE. This study aims to investigate the RAE in Luxembourg that follows an 'open-door' selection policy in youth football due to the limited pool of players. Birthdates from all licensed and actively playing Luxembourgish youth footballers including all players of the youth national teams (396 girls and 10981 boys) competing in the season 2018/2019 were analysed and categorised into birth quarters and semesters. To further investigate a performance dependence of the RAE in amateur leagues, success was determined based on the teams' rankings at the end of the season. Differences between observed and expected birthdate distributions were calculated across all licensed players and age groups, within the national teams, and for the top- and bottom-tier football teams using chi-square statistics. While a RAE was absent across all age groups (except U7), significant RAEs with high effect sizes were observed in the top-level and national teams. These findings contrast the substantial RAE effects in large football nations and suggest that open selection systems might reflect an environmental constraint that limit the prevalence of RAE in football. Further, this study indicates that a performance dependence of the RAE is not limited to high level football but already occurs on an amateur level.

## Introduction

Grouping individuals by age is a widely used policy in sport to provide equal opportunity and fair competition [1]. Athletes are allocated to cohorts based on chronological age, separated by cut-off dates. In 1997, the International Federation of Association Football (FIFA) established the 1st of January as the cut-off date in football for all youth categories. As a result, the age difference within one birth cohort can go up to one year, known as the relative age. In football, the relative age difference can increase to 24 or 36 months when age groups are defined in two or three-year bands [2]. Especially during puberty, inter-individual differences in the onset and speed of maturational growth, and consequently physiological performance [3], can vary

contacted to get permission for access to the data (FLF@football.lu). The authors were granted permission to analyse the dataset after all personal player information had been removed by the FLF. The authors were not allowed to share the dataset.

**Funding:** The author(s) received no specific funding for this work.

**Competing interests:** The authors have declared that no competing interests exist.

**Table 1. Overview of the female and male football players and the age categories in Luxembourgish youth football.**

| Age group | n | Age |
|---|---|---|
| **(1) Luxembourg** | | |
| U7 ♂ | 1224 | 5.9 ± 0.5 |
| U9 ♂ | 2264 | 7.8 ± 0.6 |
| U11 ♂ | 1839 | 9.8 ± 0.6 |
| U13 ♂ | 1825 | 11.7 ± 0.6 |
| U15 ♂ | 1520 | 13.8 ± 0.6 |
| U17 ♂ | 1257 | 15.7 ± 0.6 |
| U19 ♂ | 1052 | 17.8 ± 0.6 |
| Subtotal U7 to U19 ♂ | 10981 | |
| U15 ♀ | 299 | 12.9 ± 1.2 |
| U19 ♀ | 97 | 16.5 ± 1.1 |
| Subtotal U15 to U19 ♀ | 396 | |
| Total | 11377 | |
| **(2) Luxembourgish national teams** | | |
| U12 ♂ | 55 | 11.3 ± 0.3 |
| U13 ♂ | 39 | 12.2 ± 0.5 |
| U14 ♂ | 33 | 13.3 ± 0.3 |
| U15 ♂ | 25 | 14.3 ± 0.3 |
| U16 ♂ | 25 | 15.2 ± 0.6 |
| U17 ♂ | 14 | 16.3 ± 0.3 |
| U19 ♂ | 29 | 17.7 ± 0.6 |
| Total | 220 | |
| **(3) Top–/ Bottom– 6 teams** | | |
| U13 ♂ | 90 / 98 | 12.0 ± 0.6 / 11.7 ± 0.6 |
| U15 ♂ | 87 / 96 | 13.9 ± 0.6 / 13.7 ± 0.6 |
| U17 ♂ | 98 / 109 | 16.0 ± 0.5 / 15.6 ± 0.6 |
| U19 ♂ | 115 / 105 | 17.8 ± 0.6 / 17.7 ± 0.6 |
| Subtotal U13 to U19 ♂ | 390 / 408 | |
| U15 ♀ | 86 / 90 | 13.2 ± 1.2 / 12.7 ± 1.3 |
| Total | 476 / 498 | |

♀ = female, ♂ = male, Values present mean ± standard deviation

significantly between members within one age category [4]. As coaches tend to prefer short-term success [5], late-born players who appear to be delayed in their maturation due to a lower relative age are often mistakenly considered "less gifted" [6]. Such approach can lead to a relative age effect (RAE), which is defined as the asymmetry in the birth date distribution of players within one age category [7]. It typically reflects an overrepresentation of chronologically older participants compared to their relatively younger peers [7, 8]. Relatively younger players are often not selected by elite academies that may result in a loss of talent.

Cobley et al. [7] showed that the RAE is a robust phenomenon and particularly prevalent in football where physical factors such as power, growth and maturation provide substantial advantage for players. The RAE in football has been confirmed in many countries including Belgium [9], Germany [10], Switzerland [11], China [12] and the United States [13], suggesting its presence across diverse environments [14]. According to the developmental systems model introduced by Wattie et al. [4] the high prevalence of the RAE in football is attributable to

individual, task, and environmental constraints. An earlier birthdate relative to the cut-off date associated with a growth and maturation advantage (individual constraint) will likely result in selection advantage in sports such as football where greater size and strength are positively related to performance (task constraint).

The RAE in football is not only evident in male players but has repeatedly been reported also in the female population. A high RAE prevalence was observed in European and American regions [15] and especially at younger age (10–14 years) when compared to older players (>15 years) in the Swiss female soccer population [16]. Sedano et al. [17] further reported an RAE dependence on performance level as reflected by a significant overrepresentation of early-born female players in the national teams and the first and second but not third division of Spanish female football. These findings are well in accordance with a recent meta-analysis by Smith et al. [18] including 33 studies in female football. The authors reported a significant RAE as reflected by an overrepresentation of early-born female athletes that adhere to previous findings in male football [7]. Together, these results suggest that the individual and task constraints for male and female soccer players are comparable and result in a similar pattern of RAE prevalence across age groups and performance levels.

The popularity of football especially in many European countries results in a large pool of players that favour performance-focussed selection procedures (environmental constraint) and in turn increase the prevalence of the RAE [4]. However, while this interaction of constraints applies to many established football nations such as Germany or Spain [2, 19], research in smaller countries is limited. These nations are often forced to apply a more open system including flexible selection processes ('open-door' policy) due to the limited number of players. According to the developmental systems model this environmental constraint may reduce the risk of the RAE in smaller countries. In fact, recent investigations in Scotland [20] and Israel [5] suggest a lower prevalence of RAE. However, the samples in these studies were small and comparisons between performance levels did not consider age as a potential influencing factor. Therefore, it remains unclear if the environmental constraint of a less selective, open-door policy in smaller countries reduces the risk of RAE across age ranges and performance levels in youth football.

This study aims to quantify the prevalence of the RAE in Luxembourgish youth football. In contrast to previous research, the entire population of active male and female football players between U7-U19 rather than a sample of participants was considered. Luxembourg predestines for the RAE analysis since soccer has a similar popularity when compared to other established football nations (about 58% of all registered athletes in Luxembourg are playing soccer). However, as a small country, Luxembourg follows an 'open-door' policy without strict selection processes at amateur level aiming to minimize the drop-out of players in youth football. This study investigated the RAE across age categories from U7-U19 in male and female players as well as in the Luxembourgish youth national teams. Moreover, the relation between RAE prevalence and performance was analysed by comparing the teams finishing the season at top and bottom of the table. It was hypothesized that across all players and teams there should be no RAE in Luxembourgish football due to the less strict player selection processes. However, an RAE was expected in the national teams and top-tier football teams that have a more competitive focus associated with a higher RAE prevalence [7].

## Methods

### Participants

All 11377 registered and active youth football players (396 girls (3,48%) and 10981 boys (96,52%)) from 103 football clubs holding a licence from the Luxembourgish Football

Federation (FLF) were included in this study. Based on the players' birthdates, age categories from under- 7 (U7) to U19 for males and from U15 to U19 for female football players were defined. The age groups represented every active youth football player, every existing age category and every club in Luxembourgish football. Additionally, all Luxembourgish youth national teams, consisting of 220 male players ranging from U12 to U19, were analysed. An overview of the different age categories is presented in Table 1. Anonymised data was collected from the national database with permission from the FLF and with the consent of the players or the parents (under legal age). This study was approved by the local research committee of LUNEX University (Nr: 21_E_010) in accordance with the Declaration of Helsinki (2013). Players (or parents in case players are under legal age) registered by the Luxembourgish Football Federation (FLF) gave their written informed consent that player data can be used for research purposes.

## Procedures

Based on the birthdate, each player was assigned to one of four relative age quarters (Q) and two semesters (S) in order to calculate the RAE of the different age categories [7]. In Luxembourg, the cut-off date for football is the 1$^{st}$ of January. Therefore, the year was divided into quarters: Q1 = January, February and March, Q2 = April, May and June, Q3 = July, August and September, Q4 = October, November and December. Furthermore, the year was also divided into semesters: S1 = January to June and S2 = July to December for half-year distributions. Every player having at least one game nomination throughout the 2018/19 season (last fully completed season before the COVID-19 pandemic) was included. U7 and U9 could not be sampled by game nominations because they do not play competitively in Luxembourg, and game nominations were not recorded. Therefore, every player licensed for the U7 and U9 was included in the samples. Team performance was measured based on competitive success. Competitive success was analysed comparing the team's final ranking in the championship (Top-6 teams of the first division with the Bottom-6 teams of the last division for each age category). The final ranking in the championship was established by the total points earned by the teams during the entire season. Only teams that completed every game were considered for the final ranking. Youth league structure for the season 2018/19 was determined by the FLF, reuniting 5 to 7 teams in one division for every age category (for example U17 male: Total of 61 teams, 9 divisions with 6 teams, 1 division with 7 teams). Every male age category operated over 2-year bands and 4-year bands for females. The comparison between Top and Bottom teams is based on 6 teams. The reason for this is built upon the organizational structure of the FLF, grouping an average of 6 teams together in one division. U7, U9 and U11 were not considered in the performance analysis because no competitive championship is played. The female U19 category was excluded because the total number of teams competing was insufficient (total of 6 teams). Players listed twice in championships (for example first and second team of one club) were assigned to the team in which they participated more frequently (at least 75% of total game nominations).

## Statistical analysis

Chi-squared ($\chi^2$) goodness of fit tests were used to assess differences between observed and expected birthdate distributions. The expected birthdate distributions of the Luxembourgish population were obtained from the National Institute of Statistics and Economic Studies of Luxembourg (STATEC). The birth years of interest were 2000–2007 and 2000–2013 for females and males, respectively. For the comparisons between observed and expected distributions, the observed distribution of football players was compared to the equivalent distribution

in the population matching for age and gender. For instance, the observed distribution of male U15 football players (birth years 2004–2005) was compared to the expected distribution of males born in 2004–2005 based on the STATEC data. This procedure was applied to all age groups and for both genders and ensures that the observed distribution was sampled from the expected distribution. Effect sizes for 3 degrees of freedom were calculated by using Cramer's V(small: V = 0.06–0.17, medium: V = 0.18–0.29, large: V $\geq$ 0.30). Odds ratios (OR) with 95% confidence intervals (95% CI) were calculated between Q1/Q4 and S1/S2 as previously reported by (Cobley et al., 2009). The OR comparisons were interpreted as follows: OR < 1.22, 1.22 $\leq$ OR < 1.86, 1.86 $\leq$ OR < 3.00, and OR $\geq$ 3.00, as negligible, small, medium and large, respectively [21] if the confidence interval did not include 1. Crosstab-test (Pearson's $\chi^2$) determined differences in birthdate distribution between top and bottom teams. Adjusted residuals (z-scores), were transformed into $\chi^2$ test results and into Bonferroni corrected p-values [22, 23]. The effect size was determined according to Cramer's V. An alpha level of $p < 0.05$ was set for statistical significance. All statistical analyses were performed in SPSS 27.0 (Chicago, USA).

## Results

The results for the RAE across age groups is presented in Table 2. There was a statistically significant RAE for the categories U7, U9, U15 (p = 0.01), and across all players (Total) (p < 0.001). Only the U7 ($\chi^2$ (3, $N$ = 1224) = 32.05; p < 0.001; V = 0.09) revealed a small effect size, all other statistically significant age category had a negligible effect size (V $\leq$ 0.05). Players born at the beginning of the year were overrepresented compared to late-born players in the U7 (Q1 = 28.76%; Q4 = 18.38%) resulting in a small effect (OR = 1.58). This was also observed for the half-year distribution S1 vs. S2 of the U7 (OR = 1.28). Small effect sizes for the Q1-Q4 comparisons were further observed for male U9 (OR = 1.26) and U15 (OR = 1.25). Across all players there was only a negligible effect (OR = 1.20). No relative age effect was observed in female players.

Table 3 summarizes the RAE in Luxembourgish national teams. Significant effects were only observed for the U19 ($\chi^2$ (3, $N$ = 29) = 10.13; p = 0.018; V = 0.34) with a large effect size and for the total of all the national team players ($\chi^2$ (3, $N$ = 220) = 27.90; p < 0.001; V = 0.21) with a medium effect size. OR analyses for all players of the national team revealed a medium effect size for the Q1-Q4 comparison (OR = 2.78) indicating that a youth player of the national team born in Q1 is 2.78 times more likely to be selected for the national teams than teammates born in Q4. Except the U19, all other age groups failed to reach the significance levels although effect sizes based on Cramer's V and OR revealed medium to high effect sizes. The distribution across birth quarters for the whole Luxembourgish football federation and the national teams is illustrated in Fig 1.

Analyses of the RAE in top and bottom teams of the Luxembourgish youth leagues are presented in Table 4. Statistically significant RAE effects were observed across all male top teams (p $\leq$ 0.037) as well as the total of all top team players (p < 0.001) accompanied by small to medium effect sizes. Top team players born in Q1 were overrepresented compared to Q4 in every age category for male and female ranging from (Q1 = 29.59% - 44.83% to Q4 = 14.29% - 23.48%). Odds Ratios with a medium effect size were found between Q1 and Q4 in the U13 (OR = 2.39) and Subtotal (OR = 1.91) and with a small effect size for the Total (OR = 1.78) of the top teams. Also half-year distributions were significant in the top teams, associated with medium (OR = 1.89–1.95) and small (OR = 1.67–1.80) effect sizes, except for the male U19. RAE was absent in the female top U15 teams (p = 0.022) as well as all male (p $\geq$ 0.261) and female (p = 0.086) bottom teams. The player distribution across birth quarters in the male and female top and bottom teams is presented in Fig 2.

**Table 2. Relative age effect analyses across age groups in Luxembourgish youth football.**

| Age group | N | Birthdate distribution (%) | | | | $\chi^2$ | p | V | Odds Ratio (95% CI) | |
| | | Q1 | Q2 | Q3 | Q4 | (df = 3) | | | Q1 vs. Q4 | S1 vs. S2 |
|---|---|---|---|---|---|---|---|---|---|---|
| U7 ♂ | 1224 | 352 (28.76) | 330 (26.96) | 317 (25.90) | 225 (18.38) | 32.05 | **0.001** | 0.09 | 1.58 (1.25–1.99)* | 1.28 (1.09–1.50)* |
| U9 ♂ | 2264 | 630 (27.83) | 585 (25.84) | 565 (24.96) | 484 (21.38) | 17.88 | **0.001** | 0.05 | 1.26 (1.07–1.49)* | 1.18 (1.05–1.33)* |
| U11 ♂ | 1839 | 490 (26.64) | 438 (23.82) | 472 (25.67) | 439 (23.87) | 2.35 | 0.503 | 0.02 | 1.05 (0.88–1.26) | 1.04 (0.91–1.18) |
| U13 ♂ | 1825 | 445 (24.38) | 461 (25.26) | 493 (27.01) | 426 (23.34) | 4.85 | 0.183 | 0.03 | 1.09 (0.90–1.31) | 1.03 (0.91–1.17) |
| U15 ♂ | 1520 | 433 (28.49) | 362 (23.82) | 398 (26.18) | 327 (21.51) | 11.30 | **0.010** | 0.05 | 1.25 (1.02–1.53)* | 1.08 (0.93–1.24) |
| U17 ♂ | 1257 | 334 (26.57) | 331 (26.33) | 305 (24.26) | 287 (22.83) | 5.14 | 0.162 | 0.04 | 1.15 (0.92–1.44) | 1.13 (0.97–1.32) |
| U19 ♂ | 1052 | 290 (27.57) | 269 (25.57) | 256 (24.33) | 237 (22.53) | 6.57 | 0.087 | 0.05 | 1.18 (0.92–1.51) | 1.11 (0.94–1.32) |
| U15 ♀ | 299 | 81 (27.09) | 77 (25.75) | 66 (22.07) | 75 (25.08) | 3.07 | 0.382 | 0.06 | 1.10 (0.70–1.72) | 1.15 (0.83–1.59) |
| U19 ♀ | 97 | 20 (20.62) | 24 (24.74) | 27 (27.84) | 26 (26.80) | 1.25 | 1.248 | 0.07 | 0.77 (0.34–1.73) | 0.81 (0.46–1.43) |
| Total | 11377 | 3075 (26.44) | 2877 (25.34) | 2899 (25.36) | 2526 (22.86) | 45.43 | **0.001** | 0.04 | 1.20 (1.11–1.29)* | 1.11 (1.05–1.16)* |

♀ = female; ♂ = male; Q1-4, quarter; S1-2, semester; $\chi^2$, Chi-squared; df, degrees of freedom; p, significance

V, Cramer's V; RP, reference population; Bold = Significant at an alpha of p < 0.05

* Significant OR

Differences in birthdate distribution between the Top-6 and Bottom-6 teams is presented in Table 5. A statistical significance was found in the initial Pearson's chi-squared test only for male ($\chi^2$ (3, N = 798) = 19.698; p < 0.001; V = 0.157) but not female players ($\chi^2$ (3, N = 176) = 1.285; p = 0.733). The adjusted residuals (z-scores) between birth quarters were transformed into Bonferroni corrected chi-squared results and showed significant results in Q1 ($\chi^2$ = 10.23; p = 0.006; V = 0.065) and Q3 ($\chi^2$ = 11.45; p = 0.003; V = 0.069) with a small effect size for male top and bottom players.

## Discussion

This study provides a comprehensive analysis of the relative age effect (RAE) across age groups, gender and performance levels including the whole population of active Luxembourgish football players. In line with the hypotheses, there were negligible (if at all) RAEs except for

**Table 3. Relative age effect analyses across Luxembourgish national teams.**

| ♂ | | Birthdate distribution (%) | | | | $\chi^2$ | p | V | Odds Ratio (95% CI) | |
| Age group | N | Q1 | Q2 | Q3 | Q4 | (df = 3) | | | Q1 vs. Q4 | S1 vs. S2 |
|---|---|---|---|---|---|---|---|---|---|---|
| U12 | 55 | 17 (30.91) | 16 (29.09) | 16 (29.09) | 6 (10.91) | 6.64 | 0.084 | 0.20 | 3.05 (0.92–10.11) | 1.62 (0.76–3.45) |
| U13 | 39 | 16 (41.03) | 10 (25.64) | 6 (15.38) | 7 (17.95) | 6.21 | 0.102 | 0.23 | 2.29 (0.66–7.96) | 2.11 (0.84–5.26) |
| U14 | 33 | 13 (39.39) | 9 (27.27) | 6 (18.18) | 5 (15.15) | 3.48 | 0.324 | 0.19 | 2.31 (0.57–9.41) | 1.78 (0.66–4.78) |
| U15 | 25 | 8 (32.00) | 9 (36.00) | 4 (16.00) | 4 (16.00) | 4.12 | 0.249 | 0.23 | 2.00 (0.38–10.41) | 2.30 (0.73–7.27) |
| U16 | 25 | 9 (36.00) | 10 (40.00) | 5 (20.00) | 1 (4.00) | 7.12 | 0.068 | 0.31 | 9.00 (0.85–94.9) | 2.92 (0.87–9.78) |
| U17 | 14 | 5 (35.71) | 3 (21.43) | 5 (35.71) | 1 (7.14) | 2.64 | 0.451 | 0.25 | 5.00 (0.34–72.77) | 1.56 (0.34–7.11) |
| U19 | 29 | 14 (48.28) | 5 (17.24) | 5 (17.24) | 5 (17.24) | 10.13 | **0.018** | 0.34 | 2.80 (0.65–12.09) | 1.90 (0.67–5.42) |
| Total | 220 | 82 (37.62) | 62 (28.10) | 47 (21.66) | 29 (12.63) | 27.90 | **0.001** | 0.21 | 2.78 (1.57–4.90)* | 1.91 (1.30–2.81)* |

♂ = male; Q1-4, quarter; S1-2, semester; $\chi^2$, Chi-squared; df, degrees of freedom; p, significance

V, Cramer's V; Bold = Significant at an alpha of p < 0.05

* Significant OR

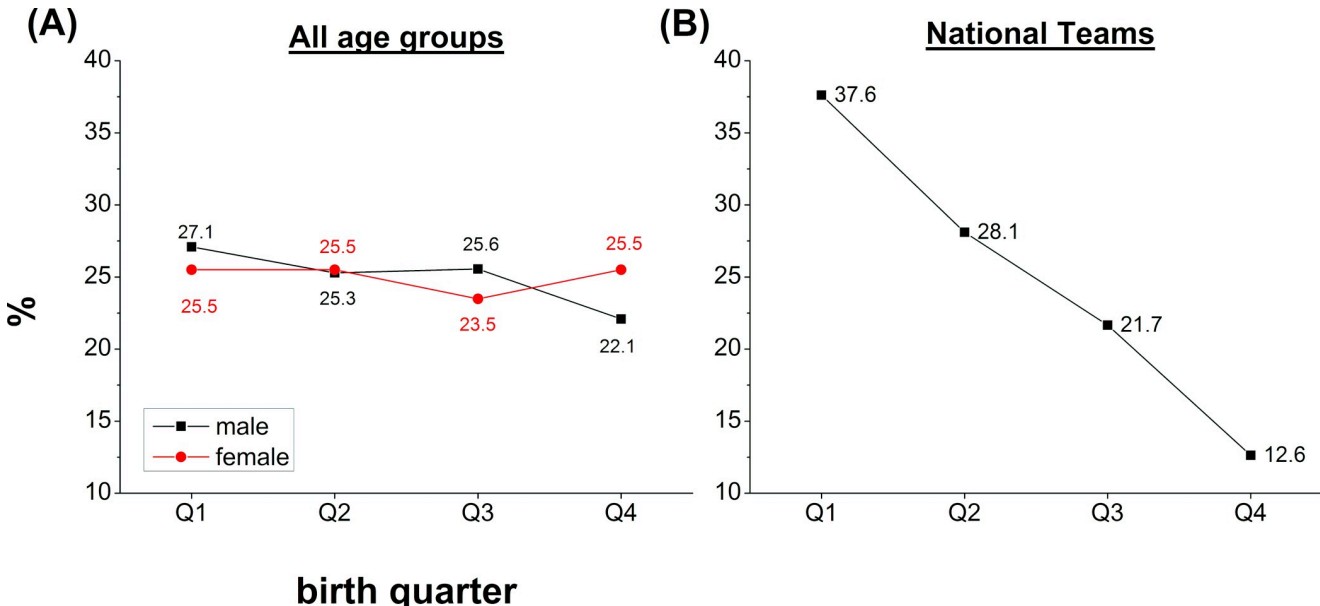

**Fig 1. Male and female player distribution across age groups.** (A) Distribution of male (black) and female (red) players in the whole population of Luxembourgish football players. (B) Player distribution across birth quarters in the Luxembourgish national teams. Data points reflect average values across age groups.

the U7 group that indicated a small effect size. Top teams revealed significantly higher RAEs when compared to bottom tier clubs indicating an effect of competitive level. This was further supported by the large RAE in the Luxembourgish national teams. These findings suggest that the 'open-door' policy in Luxembourgish football associated with less strict selection criteria

**Table 4. Relative age effect analyses of the top-6 and bottom-6 teams.**

| Age group | N | Birthdate distribution (%) | | | | $\chi^2$ (df = 3) | p | V | Odds Ratio (95% CI) | |
|---|---|---|---|---|---|---|---|---|---|---|
| | | Q1 | Q2 | Q3 | Q4 | | | | Q1 vs. Q4 | S1 vs. S2 |
| Top– 6 teams | | | | | | | | | | |
| U13 ♂ | 90 | 32 (35.56) | 26 (28.89) | 18 (20.00) | 14 (15.56) | 9.88 | **0.020** | 0.19 | 2.39 (1.01–5.64)* | 1.89 (1.04–3.44)* |
| U15 ♂ | 87 | 39 (44.83) | 19 (21.84) | 11 (12.64) | 18 (20.69) | 19.47 | **0.001** | 0.27 | 2.07 (0.91–4.69) | 1.95 (1.06–3.61)* |
| U17 ♂ | 98 | 29 (29.59) | 34 (34.69) | 21 (21.43) | 14 (14.29) | 9.09 | **0.028** | 0.18 | 2.07 (0.88–4.86) | 1.80 (1.02–3.19)* |
| U19 ♂ | 115 | 38 (33.04) | 32 (27.83) | 18 (15.65) | 27 (23.48) | 8.51 | **0.037** | 0.16 | 1.36 (0.66–2.80) | 1.53 (0.91–2.58) |
| Subtotal ♂ | 390 | 138 (35.38) | 111 (28.46) | 68 (17.44) | 73 (18.72) | 35.16 | **0.001** | 0.17 | 1.91 (1.28–2.85)* | 1.77 (1.33–2.35)* |
| U15 ♀ | 86 | 29 (33.72) | 20 (23.26) | 17 (19.77) | 20 (23.26) | 4.41 | 0.220 | 0.13 | 1.45 (0.63–3.33) | 1.32 (0.73–2.41) |
| Total | 476 | 167 (35.08) | 131 (27.52) | 85 (17.86) | 93 (19.54) | 37.98 | **0.001** | 0.16 | 1.78 (1.24–2.55)* | 1.67 (1.29–2.17)* |
| Bottom– 6 teams | | | | | | | | | | |
| U13 ♂ | 98 | 25 (25.51) | 29 (29.59) | 18 (18.37) | 26 (26.53) | 3.08 | 0.379 | 0.10 | 1.00 (0.46–2.19) | 1.28 (0.73–2.24) |
| U15 ♂ | 96 | 21 (21.88) | 22 (22.92) | 28 (29.17) | 25 (26.04) | 1.08 | 0.783 | 0.06 | 0.81 (0.36–1.82) | 0.81 (0.46–1.43) |
| U17 ♂ | 109 | 33 (30.28) | 21 (19.27) | 32 (29.36) | 23 (21.10) | 4.00 | 0.261 | 0.11 | 1.38 (0.65–2.95) | 0.96 (0.57–1.64) |
| U19 ♂ | 105 | 23 (21.90) | 26 (24.76) | 34 (32.38) | 22 (20.95) | 2.34 | 0.505 | 0.09 | 1.05 (0.47–2.34) | 0.88 (0.51–1.50) |
| Subtotal ♂ | 408 | 102 (25.00) | 98 (24.02) | 112 (27.45) | 96 (23.53) | 0.60 | 0.897 | 0.02 | 1.04 (0.70–1.54) | 0.96 (0.73–1.27) |
| U15 ♀ | 90 | 28 (31.11) | 27 (30.00) | 14 (15.56) | 21 (23.33) | 6.61 | 0.086 | 0.16 | 1.33 (0.59–3.02) | 1.57 (0.87–2.84) |
| Total | 498 | 130 (26.10) | 125 (25.10) | 126 (25.30) | 117 (23.49) | 0.60 | 0.896 | 0.02 | 1.09 (0.77–1.56) | 1.06 (0.83–1.36) |

♀ = female; ♂ = male; Q1-4, quarter; S1-2, semester; $\chi^2$, Chi-squared; df, degrees of freedom; p, significance

V, Cramer's V; Bold = Significant at an alpha of p < 0.05; * Significant OR

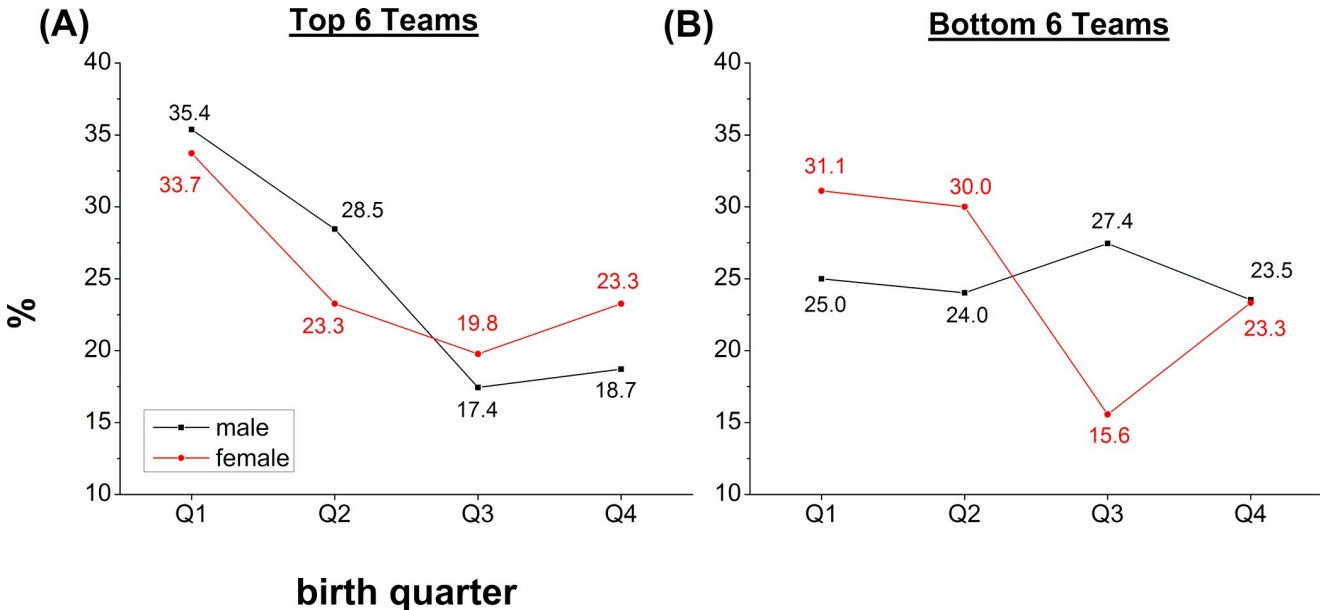

**Fig 2. Male and female player distribution for top and bottom teams.** Distribution of football male (black) and female (red) players across birth quarters in the top (A) and bottom (B) teams of the Luxemburgish football leagues. Data points reflect average values across age groups.

reduced the prevalence of the RAE in youth teams. In contrast, more performance oriented top-tier and national teams show systematic RAEs that are comparable to established football nations. According to the developmental systems model, the less strict selection policy in small countries can be considered an environmental constraint reducing the RAE prevalence.

### RAE across age groups

Only three (U7, U9, U15) age categories revealed a significant RAE that were however accompanied by negligible effect sizes, except for the U7. Significant findings were thus likely the

**Table 5. Comparison of birthdate distribution between top-6 and bottom-6 teams.**

| Performance* Quarter Crosstabulation | | | | | Quarter | | | | Total |
|---|---|---|---|---|---|---|---|---|---|
| ♂ | | | | | Q1 | Q2 | Q3 | Q4 | |
| Performance | | Top | | Count | 138 | 111 | 68 | 73 | 390 |
| | | | | % within Performance | 35.40% | 28.50% | 17.40% | 18.70% | 100.00% |
| | | | | Adjusted Residual | 3.198 | 1.427 | -3.384 | -1.663 | |
| | | Bottom | | Count | 102 | 98 | 112 | 96 | 408 |
| | | | | % within Performance | 25.00% | 24.00% | 27.50% | 23.50% | 100.00% |
| | | | | Adjusted Residual | -3.198 | -1.427 | 3.384 | 1.663 | |
| Total | | | | Count | 240 | 209 | 180 | 169 | 798 |
| | | | | % within Performance | 30.10% | 26.20% | 22.60% | 21.20% | 100.00% |
| Statistics | | z-score | | | 3.198 | 1.427 | -3.384 | -1.663 | |
| | | $\chi^2$ | | | 10.23 | 2.04 | 11.45 | 2.77 | |
| | | p | | | **0.006** | 0.614 | **0.003** | 0.385 | |
| | | V | | | 0.065 | 0.029 | 0.069 | 0.034 | |

♂ = male; Q1-4, quarter; $\chi^2$, Chi-squared; p, significance; V, Cramer's V

Bold = Significant at an alpha of p < 0.05

result of the large sample rather than a systematic RAE in youth Luxembourgish football. In the same vein, the two age categories of female players did not a reveal significant RAE. These findings contrast previous research in established football nations suggesting a RAE in youth football for both male and female populations [10, 16, 17, 24–26]. However, these nations benefit from a large athlete pool that allow performance-oriented selection procedures. In contrast, Luxembourg applies an open-door policy to account for the limited number of players. The absence of the RAE in Luxembourgish football is thus likely attributable to the less strict selection procedures. This is further supported by results from countries such as Scotland [20] and Israel [5] that apply similar player selection policies and likewise show less RAE prevalence. The open-door policy in Luxembourg and other small nations encourage children to join a sports program and continue their sports experience. The main aim is not to lose players since the overall number is limited due to the small Luxembourgish population. In 2018 only 78532 children in the age groups considered in this study were living in Luxembourg. Consequently, coaches are somewhat obliged to keep all their players and avoid selecting only the best. Thus, late-born players remain active club members and have the same playing and training opportunities as their early-born counterparts. While this policy is suggested to negatively affect individual and team performance at younger age [27], it may benefit success in the long-term. Analyses on the impact of the RAE on performance in fielding games such as football indicate positive short-term effects on team success especially at younger age (<19 years) but an association with the RAE reversal at adult level [27]. It has been suggested that relatively young players have to overcome the early stages of their career due to anthropometric/physiological disadvantages which could result in the development of cognitive and psychological abilities that benefit performance at older age [28, 29]. Recent data on the market value of football players indicating higher values for late born players at adult level support this hypothesis [30]. Therefore, the open-door player selection policy in Luxembourg may counteract the often-criticized loss of talent associated with the RAE and could thus benefit individual and team performance in the long term. As such, the open-door policy could be considered an environmental constraint limiting performance-focussed selection [6, 20] and reducing the prevalence of the RAE. However, as indicated by the RAE prevalence in national and top football teams (see below), the open-door policy is likely forced by the limited pool of football players, rather than a strategy to improve long-term success in Luxembourgish football.

Another factor contributing to the missing RAE in this study may be the concept of 'depth of competition' [31] that especially applies to small countries such as Luxembourg. This hypothesis is comparable to a lack of competition with a universal enrolment of all the players at the same level [17]. This policy aims to provide equal opportunities for children without the pressure of competition. Coaches are forced to be more flexible which might result in a lower prevalence of RAE. However, more studies in smaller countries and analyses in amateur youth football are needed to support this hypothesis. Together these findings fit well in the developmental systems model introduced by Wattie et al. (2015). The lower number of players in small countries associated with an open-door selection policy may represent an environmental constraint that reduces the RAE prevalence in football.

The only significant RAE that exceeded a negligible effect size was observed for the U7 category. This may be surprising as the RAE is considered being most prevalent during puberty (12–16 years), where the biggest physical differences occur due to the maturation process [32, 33]. A potential explanation may be self-selection processes. Independent from selection by coaches, early born children might self-select earlier for playing organized football due to physical advantages. They might overcome the first hurdle and initiate earlier in age group participation in a football club than their younger counterparts which may result in the small RAE observed in this study.

For the female population, no RAE was observed in any of the age groups. Further, the effect sizes were slightly lower when compared to the male athletes. Although this may be in line with the lower performance-enhancing effect of the RAE in female when compared to male athletes [27] it contrast previous studies and meta analyses suggesting RAE effects not only in males but also female football players [16–18]. Similar to male football, this discrepancy is best explained by the low number of female players in Luxembourg (<400 in 2018) that limit the options for player selection based on performance criteria. The suggested obligation of coaches to keep players in the male teams may be even more pronounced for female football thus reducing the RAE prevalence. The low number of players further resulted in the creation of 4-year age categories for female football when compared to 2-year categories for males. Consequently, the relative age may lose relevance for performance in broad age categories which manifests in the missing RAE in Luxembourgish female football.

## RAE and performance

The overall RAE for all Luxembourgish national team players showed that players born at the beginning of the year were consistently over-represented. These results are well in line with several studies reporting significant RAEs in high-level youth football [6, 10, 34]. Effects sizes were comparable between age groups, although there was a trend towards greater effect sizes at higher age. While the RAE is generally expected to decrease with age and especially after puberty [19, 20, 24] since technical, tactical and psychological skills become more important for selection processes, this can be explained by the cascade effect [20, 35]. The cascade effect suggests a continuation of bias favouring older players over younger ones due to several years of (de)selection processes affected by the RAE. Players were able to benefit from their physical and psychological advantages over several years of selection processes affected by coaches' decisions that results in the large RAE in the U19 group.

The results of the national teams support the higher prevalence of RAEs in high-level and more performance-oriented teams [26]. When compared to the open-door policy at lower levels, stricter selection processes in national teams favour more mature and physically developed players that result in the observed RAE. Therefore, while the Luxembourgish selection policy at amateur level may be more liberal and less performance-oriented when compared to other countries, the focus on success in national teams results in a similar RAE on high performance level. This is further supported by the RAE results for the Top-6 and Bottom-6 teams of the Luxembourgish football leagues. Although all teams play on an amateur level, the RAE was prevalent in all top teams but was observed in none of the lower ranked teams. Further, there was a significant difference in birthdate distribution between top and bottom teams indicating a higher proportion of Q1 players in the top teams, while at the bottom of the table, teams have more players from Q3. These findings extend previous research on the difference between high-level and amateur teams [6, 20] and suggest that even on the amateur level there are differences in the prevalence of the RAE. While a relation between RAE and performance has previously been shown in young alpine skiers [36] especially in football research on the relation between the RAE and sport-specific performance is limited and other studies such as Kirkendall [37] were not able to find a significant relation between RAE and team performance. Nonetheless, the consistent pattern of results indicating a significant RAE for all top-level but not bottom-level teams add further support to the performance dependence of the RAE even in amateur football.

In contrast to males, top teams in Luxembourgish female football did not exhibit a significant RAE. This may again be explained by a lower performance effect of the RAE in female when compared to male football [27]. It however contrasts a recent meta-analysis suggesting

an increasing RAE prevalence in female soccer with increasing performance level [18]. As such, an RAE may be expected in female top teams, however comparisons between age quarters did not reach significance and effect sizes were comparable between the top and bottom teams. These findings may again be explained by the very limited number of players that substantially limit player performance-based selection when compared to male football teams. The broad age categories of 4 years may have further reduced the effects of relative age and resulted in an even distribution of players across birth quarters. With the increasing popularity of female football, future studies will have to monitor the development of the RAE in female soccer teams during the next years.

## Limitations

Since maturation of performance benefits in early-born players are suggested to link the RAE prevalence to sport performance and competitive success, future research should aim to measure anthropometric and physical performance as additional outcome variables. Further, low sample sizes for the female groups and age categories that span four years, may mask a potential RAE. Therefore, results on the RAE in the female players must be interpreted with caution. Future studies may further apply longitudinal approaches to identify the development of the RAE across age levels by not only comparing to birth distribution of the general population but the active football players [8, 38].

## Conclusion

The results indicate negligible relative age effects in Luxembourgish youth football. This can be explained by the open-door player selection criteria applied in small countries which can be considered as an environmental constraint reducing the RAE prevalence. Nonetheless, higher performing Luxembourgish clubs and national teams were still characterized by a considerable RAE. These findings support the dependence of the RAE on the competitive level and suggest performance-oriented selection processes even in small countries where football is performed on an amateur level.

## Supporting information

**S1 File. Inclusivity in global research.**
(DOCX)

## Author Contributions

**Conceptualization:** Claude Simon, Thorben Hülsdünker.

**Data curation:** Claude Simon.

**Formal analysis:** Claude Simon.

**Investigation:** Claude Simon.

**Methodology:** Claude Simon, Thorben Hülsdünker.

**Project administration:** Claude Simon, Fraser Carson, Thorben Hülsdünker.

**Resources:** Thorben Hülsdünker.

**Software:** Thorben Hülsdünker.

**Supervision:** Fraser Carson, Thorben Hülsdünker.

**Validation:** Claude Simon.

**Visualization:** Claude Simon.

**Writing – original draft:** Claude Simon, Fraser Carson, Irene Renate Faber, Thorben Hülsdünker.

**Writing – review & editing:** Claude Simon, Fraser Carson, Irene Renate Faber, Thorben Hülsdünker.

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
