## [Decision Letter · Decision Letter 0]

16 May 2022

PONE-D-22-09363Luxembourg’s open-door player selection policy reduces the risk of a relative age effect in youth footballPLOS ONE

Dear Dr. Hülsdünker,

Thank you for submitting your manuscript to PLOS ONE. After careful consideration, we feel that it has merit and we invite you to make a minor revisions to meet PLOS ONE’s publication criteria. Therefore, we invite you to submit a revised version of the manuscript that addresses the points raised during the review process.

We look forward to receiving your revised manuscript.

Kind regards,

Rafael Franco Soares Oliveira

Academic Editor

PLOS ONE

Journal Requirements:

Reviewers' comments:

Reviewer's Responses to Questions

**Comments to the Author**

1. Is the manuscript technically sound, and do the data support the conclusions?

Reviewer #1: Partly

Reviewer #2: Yes

2. Has the statistical analysis been performed appropriately and rigorously? 

Reviewer #1: Yes

Reviewer #2: Yes

3. Have the authors made all data underlying the findings in their manuscript fully available?

Reviewer #1: No

Reviewer #2: Yes

4. Is the manuscript presented in an intelligible fashion and written in standard English?

Reviewer #1: Yes

Reviewer #2: Yes

5. Review Comments to the Author

Reviewer #1: The authors' research question is well defined and supported in the statistical analyses presented.

The study does not prove the identified cause for the inexistence of RAE in Luxembourgish youth football. It only proves the non-existence of RAE, not the cause, as suggested by the title of the article: "Luxembourg’s open-door player selection policy reduces the risk of a relative age effect in youth football".

The only suggestion I would make to the paper is to introduce 1 or 2 charts showing the results, as they make the paper more appealing.

To make the title of the paper truly match the results it should be something like: " Luxembourg's youth football teams exhibit a negligible relative age effect".

Reviewer #2: Thank you for the opportunity to review this article. The paper addresses a novel under-researched are. However, there are some questions that need to be addressed to the manuscript.

KEYWORDS

Please, avoid using the same words of the title in the keywords.

INTRODUCTION

The introduction section is clear and well-written. However, it is necessary to elaborate a paragraph explaining in depth the relationship between RAE and performance in both genders.

MATERIAL AND METHODS

Name “under” before. (line 97)

Add the year of the Declaration of Helsinki (line 104)

Are these years correct? (line 141)

“p” in italic (line 157)

DISCUSSION

The discussion is clear and consistent. Great job! However, where the discussion of female results? Please, include it.

Relative age effects need to be abbreviated. (line 207)

Add population (line 233)

What will happen to these talented players if they cannot enhance their performance for this reason? Maybe, they would not be able to be successful in the future? This idea is not good for the reader... Please, discuss it. (line 233-234)

6. PLOS authors have the option to publish the peer review history of their article (what does this mean?). If published, this will include your full peer review and any attached files.

Reviewer #1: No

Reviewer #2: No

---

## [Author Response · Author response to Decision Letter 0]

16 Jul 2022

1) Title page has been adjusted according to the formatting guidelines although the placement of "#" is not clear. Formatting for headlines, figures and tables has been changed according to the guidelines. 

2) Questionnaire is part of the re-submission

3) This study used a third-party dataset that we are not allowed to make publicly available. The data availability statement has been changed and contact information to request the data from the Luxembourgish Football Federation is provided

4) The reference list has been updated and new references were included during the revision process

The point-by-point answers to the reviewer comments are included in the corresponding file as part of the re-submission

Note: Please respond to my new university mail address thorben.huelsduenker@lunex-university.net. I am not able to change the mail address since PLOSOne assigned another account to this address for a review. I already sent several mails about this issue so probably it is already in progress. 

However, if there are any problems with the manuscript please send any information to my new mail account mentioned above, otherwise I will not have access. 

Thank you for your support

---

## [Decision Letter · Decision Letter 1]

1 Aug 2022

Low prevalence of relative age effects in Luxembourg’s male and female youth football

PONE-D-22-09363R1

Dear Dr.  Hülsdünker,

We’re pleased to inform you that your manuscript has been judged scientifically suitable for publication and will be formally accepted for publication once it meets all outstanding technical requirements.

Kind regards,

Rafael Franco Soares Oliveira

Academic Editor

PLOS ONE

Additional Editor Comments (optional):

Congratulation to all authors. My recommendation is to accept.

Best regards

Reviewers' comments:

Reviewer's Responses to Questions

**Comments to the Author**

1. If the authors have adequately addressed your comments raised in a previous round of review and you feel that this manuscript is now acceptable for publication, you may indicate that here to bypass the “Comments to the Author” section, enter your conflict of interest statement in the “Confidential to Editor” section, and submit your "Accept" recommendation.

Reviewer #1: All comments have been addressed

Reviewer #2: All comments have been addressed

2. Is the manuscript technically sound, and do the data support the conclusions?

Reviewer #1: Yes

Reviewer #2: Yes

3. Has the statistical analysis been performed appropriately and rigorously? 

Reviewer #1: Yes

Reviewer #2: Yes

4. Have the authors made all data underlying the findings in their manuscript fully available?

Reviewer #1: Yes

Reviewer #2: Yes

5. Is the manuscript presented in an intelligible fashion and written in standard English?

Reviewer #1: Yes

Reviewer #2: Yes

6. Review Comments to the Author

Reviewer #1: This is a study that characterizes the reality of Luxembourg's youth football teams regarding the phenomenon of RAE, based on the age of the athletes, namely if they were born in the first or second semester of the respective year. It is a comprehensive study in terms of age groups from U7 to U19.

The phenomenon according to the authors is more noticeable as the levels of competition increase, which is natural, so here the conclusion they reach is not a novelty, but corroborates the existing literature. It would be interesting in future studies for the authors to try to discover how to overcome this phenomenon.

Reviewer #2: Congratulations on a much improved manuscript

7. PLOS authors have the option to publish the peer review history of their article (what does this mean?). If published, this will include your full peer review and any attached files.

Reviewer #1: **Yes: **Antonio Trigo

Reviewer #2: No

---

## [Editor Report · Acceptance letter]

5 Aug 2022

PONE-D-22-09363R1 

Low prevalence of relative age effects in Luxembourg’s male and female youth football 

Dear Dr. Hülsdünker:

I'm pleased to inform you that your manuscript has been deemed suitable for publication in PLOS ONE. Congratulations! Your manuscript is now with our production department. 

Kind regards, 

on behalf of

Dr. Rafael Franco Soares Oliveira 

Academic Editor

PLOS ONE